# The Relationship between Habitual Coffee Drinking and the Prevalence of Metabolic Syndrome in Taiwanese Adults: Evidence from the Taiwan Biobank Database

**DOI:** 10.3390/nu14091867

**Published:** 2022-04-29

**Authors:** Meng-Ying Lu, Hsiao-Yang Cheng, Jerry Cheng-Yen Lai, Shaw-Ji Chen

**Affiliations:** 1Division of Cardiology, Department of Internal Medicine, Taitung MacKay Memorial Hospital, Taitung 95054, Taiwan; davielu.tw@yahoo.com.com (M.-Y.L.); sheephome520@yahoo.com.tw (H.-Y.C.); 2Department of Medicine, MacKay Medical College, New Taipei City 25245, Taiwan; 3Master Program in Biomedicine, College of Science and Engineering, National Taitung University, Taitung 95092, Taiwan; chengyen@hotmail.com; 4Department of Medical Research, Taitung MacKay Memorial Hospital, Taitung 95054, Taiwan; 5Department of Psychiatry, Taitung MacKay Memorial Hospital, Taitung 95054, Taiwan

**Keywords:** coffee, black coffee, gender, metabolic syndrome, Taiwan Biobank

## Abstract

Previous studies revealed inconsistent results between coffee drinking and metabolic syndrome (MetS). The aim of the study was to evaluate the relationship between habitual coffee drinking and the prevalence of MetS among men and women. We conducted a nationwide, cross-sectional study using 23,073 adults obtained from the Taiwan Biobank database (mean ± SD (range) age, 54.57 ± 0.07 (30–79) years; 8341 men and 14,731 (63.8%) women). Adults who drank more than one cup of coffee per day (*n* = 5118) and those who drank less than one cup per day (*n* = 4515) were compared with nondrinkers (*n* = 13,439). Multivariate logistic regression models were used to evaluate the risk of MetS between the two groups. Separate models were also estimated for sex-stratified and habitual coffee-type-stratified (black coffee (BC), coffee with creamer (CC), and coffee with milk (CM)) subgroup analyses. The MetS diagnosis was based on at least three of the five metabolic abnormalities. Coffee drinkers (≥1 cup/day) had a significantly lower prevalence of MetS than nondrinkers (AOR (95% CI): 0.80 (0.73–0.87)). Women who drank any amount of coffee and any type of coffee were more likely to have a significantly lower prevalence of MetS than nondrinkers. Only men who drank more than one cup of coffee per day or black coffee drinkers were more likely to have a lower prevalence of MetS. Our study results indicate that adults with habitual coffee drinking behaviors of more than one cup per day were associated with a lower prevalence of MetS. Moreover, women could benefit from habitual coffee drinking of all three coffee types, whereas men could only benefit from drinking BC.

## 1. Introduction

Coffee has become increasingly popular in Taiwan in the past few decades. Coffee contains nearly 1000 components, such as caffeine, phenols (including chlorogenic acid and caffeic acid), and diterpenes (comprising cafestol, kahweol, lactones, niacin, and trigonelline) [1,2,3]. Many studies have proven that coffee consumption has a positive effect on chronic diseases. Habitual coffee drinking reduced arterial stiffness, blood pressure, and the incidence of new-onset hypertension [4,5,6], had a beneficial effect on weight loss [7,8,9], reduced intestinal glucose absorption [10], and decreased the incidence of type 2 diabetes mellitus [11,12]. Metabolic syndrome (MetS) is defined as a metabolic disorder, which is characterized by high fasting blood glucose, abdominal obesity, abnormal high-density lipoprotein cholesterol (HDL-C), high triglycerides, and high blood pressure, hallmarks of insulin resistance. Individuals with MetS have a high risk of stroke and coronary heart disease [13,14]. According to the survey of Nutrition and Health Surveys in Taiwan (NAHSIT), conducted from 1993–1996 and 2005–2008, a large increase in MetS prevalence was observed, from 13.6% to 25.5% [15]. Although coffee contains many components that are beneficial for individuals with chronic disease, previous studies of the relationship between habitual coffee drinking and MetS have generated inconsistent results, even more so among men and women. The inconsistent results might have been caused by different study designs. Some studies focused on the influence of daily coffee intake volume on the prevalence of MetS, whereas some studies focused on the habitual coffee type. Furthermore, some study results were generated from a small sample size or a regional area, which cannot be generalized [16,17,18,19,20,21,22,23,24,25,26,27,28,29,30,31].

This study was conducted with an aim to evaluate the relationship between habitual coffee drinking, particularly with regard to daily intake volume and habitual coffee type, and the prevalence of MetS in the Taiwanese men and women through the Taiwan Biobank database.

## 2. Materials and Methods

### 2.1. Study Population 

The Taiwan Biobank (TWB), which is sponsored by the Taiwanese government, was established to collect lifestyle and genetic information from the Taiwanese people [32,33]. The TWB is a population-based dataset designed to recruit 200,000 community-based healthy participants in the age group of 30–79 years. In 2020, 29 recruitment centers contributed information on 149,280 participants who voluntarily shared their data with the TWB. In addition to blood samples and physical examination, each participant completed a structured questionnaire on personal information and lifestyle factors during a face-to-face interview with a TWB researcher. In this study, data collected from 23,086 adult Taiwanese individuals (8347 men and 14,739 women), aged 30–79 years, were obtained from the Taiwan Biobank. All enrollees of the biobank are required to provide written informed consent prior to data collection. Ethical approval for this study was obtained from the institutional review board (IRB) of Mackay Memorial Hospital, Taipei, Taiwan (21MMHIS351e).

### 2.2. Definition of Metabolic Syndrome

The waist circumference was determined through physical examination. Fasting blood sugar, serum creatinine, HDL-C, low-density lipoprotein cholesterol (LDL-C), and triglyceride levels were determined through biochemical examinations. According to the modified National Cholesterol Education Program Adult Treatment Panel III (NCEP ATP III) definition, MetS is diagnosed if at least three of the following five metabolic abnormalities are present: (1) waist circumference ≥90 cm for men and ≥80 cm for women; (2) triglycerides ≥150 mg/dL or pharmacotherapy for hyperlipidemia; (3) HDL cholesterol <40 mg/dL for men and <50 mg/dL for women; (4) blood pressure ≥130/85 mmHg or pharmacotherapy for hypertension; (5) fasting blood glucose ≥100 mg/dL or taking glucose-lowering medication [34].

### 2.3. Habitual Coffee Drinking

The habitual coffee drinking questionnaire included daily coffee intake volume and habitual coffee type. Participants were defined as coffee drinkers if they drank coffee three or more times per week [35]. On the basis of the mean daily intake volume (where one cup of coffee = 236.59 mL (8 fluid ounces)) [36], the coffee-drinking behavior of the participants was classified as follows: no habitual coffee drinking, less than one cup of coffee per day, and at least one cup of coffee per day. Habitual coffee type included three types of coffee (black coffee, coffee with creamer, and coffee with milk), which are common coffee types consumed by the Taiwanese people. Black coffee was defined as coffee powder or extracts, potentially with added water, but no creamer or milk. Coffee with creamer and coffee with milk were defined as black coffee with creamer and milk added, respectively. Each participant who was defined as coffee drinker recorded the coffee type. The participant who had more than one habitual coffee type was categorized as “other”.

### 2.4. Other Covariates

Data on age, sex, education level, marital status, occupational status, residential area, monthly income, habitual tea drinking, alcohol consumption, regular exercise, smoking status, secondhand smoke exposure, daily sleep duration (hours), and menstrual status, were collected through questionnaires. 

The body weight and body height were determined through physical examination, and the body mass index (BMI) was calculated as the weight (in kg) divided by the height squared (in m^2^). BMI categories included underweight–normal, overweight, and obese (BMI < 24.0, 24.0–26.9, and >27.0 kg/m^2^, respectively) [37]. 

The education level was categorized as graduation from college or higher, or up to senior high-school education. Marital status was categorized as either married or other (i.e., single, divorced, separated, or widowed). Occupational status categories included employed and unemployed. Residential area categories included urban, suburban, and rural areas [38]. The monthly income-level categories were stratified as 0–20,000, 20,000–40,000, and ≥40,000 TWD per month. Daily sleep duration categories included <6 h and ≥6 h per day [39]. Vegetarians were those who maintained a daily vegetarian diet (abstaining from the consumption of red meat, poultry, seafood, insects, and the flesh of any other animal) for at least 6 months before the data collection date. Habitual tea drinkers were those who drank tea at least once a day [35].

Those who exercised for more than 150 min per week were included in the regular exercise group. Nonsmokers were those who had never smoked or had not continuously smoked for ≥6 months. Former and current smokers were those who had continuously smoked for at least 6 months but were and were not smoking, respectively, at the time of data collection. Exposure to secondhand smoke was defined as a nonsmoker who is exposed for at least 5 min/h [40]. Alcohol consumption categories included nondrinkers, former drinkers, and current drinkers (0–150 mL/week alcohol intake for 6 months, no alcohol intake for >6 months, and ≥150 mL/week alcohol intake for six consecutive months) [35]. 

### 2.5. Statistical Analysis

The characteristics of participants with different coffee-drinking behaviors were described as means and standard deviations or medians accompanied by minimum and maximum values for continuous data and as frequencies with percentages for categorical data. Each continuous variable was verified for normality of distribution by the Shapiro–Wilk test. All variables in the different groups were compared using one-way analysis of variance (ANOVA) or Kruskal–Wallis test for continuous data and chi-square tests for categorical data. Multivariable logistic regression, after adjustment for baseline demographics and lifestyle, was used to estimate the adjusted odds ratios (AORs) for the risk of MetS in men and women with different coffee-drinking behaviors. To account for sex as a potential influence on coffee-drinking behavior, separate multivariate logistic models were used for sex-stratified analysis of male and female participants. Participant characteristics with *p* < 0.05 constitute independent variables that were adjusted in the multivariable logistic model. All data transformations and statistical analyses were performed in IBM SPSS for Windows (Version 25; Armonk, NY, USA). The null hypothesis was rejected at an alpha level of 0.05.

## 3. Results

Data were obtained from the TWB database for 23,072 adult Taiwanese (mean ± SD (range) age, 54.57 ± 0.07 (30–79) years; 8341 male and 14,731 (63.8%) female, of whom 63.85% were menopausal) participants. After verifying the normality of distribution, the continuous variable (age) exhibited a nonparametric distribution. The comparison of the baseline characteristics of nondrinkers, coffee drinkers consuming <1 cup/day, and coffee drinkers consuming ≥1 cup/day is shown in Table 1. Subgroup analysis examined the relationship between sex and coffee group. This association varied significantly between men and women (*p* = 0.023 for interaction between sex and coffee group). In the analysis including all participants, nearly all parameters of general characteristics differed significantly, except for the secondhand smoke exposure (*p*-value of population, men, and women: 0.264, 0.674, and 0.185). Nearly all parameters of MetS differed significantly, except for the waist circumference (*p*-value of waist circumference, triglycerides, HDL-c, blood pressure, and fasting plasma glucose: 0.420, <0.001, <0.001, <0.001, and 0.015). In the sex-stratified analysis, no MetS parameters differed significantly for men (*p*-value of waist circumference, triglycerides, HDL-c, blood pressure, and fasting plasma glucose: 0.345, 0.098, 0.117, 0.645, and 0.430), whereas they differed significantly in women, except for waist circumference (*p*-value of waist circumference, triglycerides, HDL-c, blood pressure, and fasting plasma glucose: 0.207, <0.001, <0.001, <0.001, and 0.009). 

The unadjusted odds ratios (UORs) of daily coffee-drinking volume, personal profiles, and dietary habits in MetS are shown in Table 2. In the whole-cohort analysis, the prevalence of MetS (UOR (95% confidence interval, 95% CI): 0.82 (0.76–0.89)) in the group of coffee drinkers consuming ≥1 cup/day was significantly lower than that among coffee nondrinkers. Among women, compared with coffee nondrinkers, the prevalence of MetS in both groups of coffee drinkers (UOR (95% CI) for <1 and ≥1 cup/day: 0.90 (0.82–1.00) and 0.76 (0.69–0.83), respectively) was significantly lower. However, among men, the prevalence of MetS did not differ significantly among coffee drinkers when compared with coffee nondrinkers (UOR (95% CI) for <1 and ≥1 cup/day: 1.01 (0.89–1.14), and 0.93 (0.83–1.05), respectively). In the whole-cohort and sex-stratified analysis, the prevalence of MetS in the group of suburb and rural area had no significant difference compared to urban area except for men in the group of rural area (UOR (95% CI) for suburb and rural area of the whole population, men, and women: 1.03 (0.97–1.10), 0.95 (0.83–1.07); 1.04 (0.94–1.14), 0.82 (0.67–0.99); 1.02 (0.94–1.11), and 1.02 (0.86–1.21)). 

Table 3 presents the AORs of daily coffee volume for MetS in Taiwanese men and women in Model 1, adjusted for age, sex, and BMI, and in Model 2, which was additionally adjusted for educational level, marital status, residential area, monthly income, smoking status, alcohol consumption status, secondhand smoke exposure, habitual tea drinking, daily sleep duration, and menstrual status. In the Model 2 analysis of the whole cohort, when compared with coffee nondrinkers, the prevalence of MetS was significantly lower in both coffee-drinking groups (AOR (95% CI) for <1 and ≥1 cup/day: 0.92 (0.84–1.00) and 0.80 (0.73–0.87), respectively). Similarly, in the Model 2 analysis for women, the prevalence of MetS in both groups of coffee drinkers was significantly lower than in coffee nondrinkers (AOR (95% CI) for <1 and ≥1 cup/day: 0.86 (0.77–0.97) and 0.78 (0.69–0.87), respectively). In the Model 2 analysis of male participants, the prevalence of MetS in coffee drinkers (≥1 cup/day) differed significantly from that in coffee nondrinkers (AOR (95% CI): 0.87 (0.76–0.99)).

To evaluate which MetS parameters were influenced by coffee drinking, we separately analyzed the daily coffee intake volume for each of the five metabolic parameters in Model 2, and the results are shown in Table 3. Except for abnormal fasting glucose, every MetS parameter had a lower prevalence in coffee drinkers (≥1 cup/day) than in coffee nondrinkers (AOR (95% CI): abnormal FPG, 1.00 (0.92–1.09); abnormal waist circumference, 0.86 (0.79–0.95); abnormal triglyceride, 0.80 (0.73–0.87); abnormal HDL-C, 0.72 (0.66–0.78); abnormal BP, 0.92 (0.85–0.99)). Women, but not men, showed similar results to the overall population for the prevalence of MetS parameters in the Model 2 comparison of coffee drinkers (≥1 cup/day) and coffee nondrinkers (AOR (95% CI): abnormal waist circumference, 0.86 (0.77–0.96); abnormal triglyceride, 0.80 (0.72–0.90); abnormal HDL-C, 0.66 (0.60–0.73); abnormal BP, 0.87 (0.79–0.97); abnormal FPG, 0.97 (0.87–1.09)). In Model 2, only the prevalence of abnormal triglyceride and abnormal HDL-C levels was significantly lower in coffee nondrinkers in the analysis of the male participants (AOR (95% CI): abnormal waist circumference, 0.86 (0.76–1.05); abnormal triglyceride, 0.81 (0.72–0.92); abnormal HDL-C, 0.83 (0.72–0.95); abnormal BP, 1.01 (0.90–1.14), abnormal FPG, 1.05 (0.93–1.19)).

To evaluate the association between each type of habitual coffee drinking and MetS, the coffee drinkers which only habitually drank of black coffee, coffee with creamer, and coffee with milk were compared separately with coffee nondrinkers, and the results are shown in Table 4. Among the participants, 8877 (38.5%) drank one type of coffee, and only 756 (3.3%) drank more than one type of coffee. Moreover, 5118 (22.2%), 1328 (5.8%), and 2431 (10.5%) participants drank black coffee, coffee with creamer, or coffee with milk, respectively. In the Model 2 analysis of all three subgroups of coffee drinkers compared with coffee nondrinkers, the prevalence of MetS was significantly lower in coffee drinkers who consumed ≥1 cup of coffee/day (AOR (95% CI): black coffee, 0.80 (0.72–0.90); coffee with creamer, 0.75 (0.62–0.92); coffee with milk, 0.78 (0.67–0.91)). In the Model 2 analysis of female participants, similar results were obtained (AOR (95% CI): black coffee, 0.81 (0.69–0.94); coffee with creamer, 0.74 (0.57–0.95); coffee with milk, 0.71 (0.59–0.86)). However, in the Model 2 analysis of male participants, only the black coffee subgroup drinking ≥1 cup of coffee/day showed a significantly lower prevalence of MetS compared with coffee nondrinkers (AOR (95% CI): black coffee, 0.82 (0.69–0.98); coffee with creamer, 0.82 (0.60–1.14); coffee with milk, 1.02 (0.77–1.34)).

## 4. Discussion

According to our sex-stratified analysis, in the age group of 30–79 years, women who drank any amount of coffee and any type of coffee were more likely to have a significantly lower prevalence of MetS than nondrinkers. Furthermore, increasing daily coffee intake could yield more benefits in women. Only men who drank more than one cup of coffee per day or black coffee drinkers were more likely to have a lower prevalence of MetS. Therefore, our study results indicated that habitual coffee drinking could protect from MetS in both Taiwanese men and women, but women could benefit more than men. Among the components of MetS, both men and women (≥1 cup/day) had significantly lower prevalence of abnormal HDL-C and hypertriglyceridemia than nondrinkers, implying a potential association between habitual coffee drinking and lipid metabolism.

Worldwide, similar cross-sectional studies from Korea, Japan, Poland, Italy, and Denmark [16,17,18,19,20,21,22,23,24,25,26,27,28,29,30,31] have been conducted, with sample sizes ranging from hundreds to tens of thousands and with participants recruited from the regional to national levels. Our study data are from a large-scale, community-based cohort derived from the TWB database comprising participants enrolled since 2008 from every administrative division of Taiwan. Thus, big data from our cross-sectional study among 23,072 adult Taiwanese participants including 8341 men and 14,731 women could provide solid evidence of a factual relationship. All the information from the questionnaire was collected from the interviews at the second visit, which is more reliable than information collected only from one visit.

Our study results showed that habitual coffee drinking could protect from MetS on both men and women. There were four cross-sectional studies showing different results from those of our study. Two cross-sectional studies with approximately 500 participants recruited from a small community showed an insignificant association between coffee intake and MetS [30,31]. Kim et al. showed a positive correlation between coffee drinking and MetS, but their participants were mainly instant coffee drinkers, which differs from our study population [17]. Shin et al. enrolled only younger participants aged 30–40 years, which is a markedly different population age from that in our study, and they showed an insignificant relationship between coffee drinking and metabolic syndrome [21]. Many cross-sectional studies have shown similar results to those of our study. Grosso et al. published a Polish arm of the Health, Alcohol, and Psychosocial Factors in Eastern Europe (HAPIEE) study, which enrolled 8822 participants from Poland, and they showed a negative association between high coffee intake and MetS among men and women [18].

Most of the cross-sectional studies focused on the association between the coffee intake volume and MetS, rather than on the association between coffee types and MetS. Our study showed that women derive benefits from daily coffee intake volumes ≥1 cup/day for all three coffee types, whereas men can only derive this benefit with black coffee. A large, cross-sectional study in Korea by Kim et al. reported results from a 14,132-participant-based Health Examinees (HEXA) study, discussing the association between daily coffee intake volume and MetS, as well as between coffee types and MetS [26]. The authors obtained similar results to those of our study, wherein an inverse correlation between daily coffee intake, of both coffee types, and MetS was observed among women.

The associations between coffee and the parameters of MetS from some cross-sectional studies were inconsistent [16,18,19,29]. In our study, only the prevalence of FPG showed an insignificant difference between coffee drinkers and coffee nondrinkers. Because our study design only focused on the coffee intake volume and type, no further quantitative information about the addition of sugar, cream, and milk was obtained. Therefore, we could not determine why the results showed an insignificant correlation between coffee intake and the prevalence of abnormal fasting blood glucose. Meanwhile, the prevalence of abnormal HDL-C was significantly lower than in coffee nondrinkers in female participants, regardless of the daily coffee intake volume, which implies that coffee strongly influences the elevation of the serum HDL-C level in women. Another study that used data from the TWB revealed that coffee drinking was significantly associated with higher HDL-C levels in women, but not in men, even after adjusting for confounders including the hepatic lipase (LIPC) and cholesterol ester transfer protein (CETP) genes. Therefore, coffee drinking might be cardioprotective, especially in women [41].

There were at least three limitations of our study. First, despite a large database, our cross-sectional study design precluded the confirmation of causal associations. Second, because of the design of the questionnaire, we could not obtain quantitative information about added sugar, creamer, and milk. The questionnaire of coffee drinking types and daily intake volume from the Taiwan Biobank study referred to the “Nutrition and Health Survey in Taiwan (NAHSIT) 1993–1996: Dietary Nutrient Intakes Assessed by 24 Hour Recall” [35]; however, in this study, the questionnaire only surveyed certain dietary habits, not dietary adjustment. Therefore, it cannot be appropriately regarded as a “food frequency questionnaire”. Third, our study enrolled only participants belonging to the middle-aged to older adult population; therefore, findings from the younger population could not be included in our results.

## 5. Conclusions

In the age group of 30–79 years, coffee drinking in excess of one cup per day could reduce the prevalence of MetS among Taiwanese men and women. Women could benefit from daily coffee intake volumes of more than one cup for all three coffee types, whereas men could only obtain a benefit with black coffee.

## Figures and Tables

**Table 1 nutrients-14-01867-t001:** The general characteristics and metabolic parameters according to different daily coffee drinking volume in Taiwanese participants aged 30–79 years.

	All (*n* = 23,072)				Male (*n* = 8341)				Female (*n* = 14,731)			
	NCD	<1 cup/day	≥1 cup/day	*p*-Value	NCD	<1 cup/day	≥1 cup/day	*p*-Value	NCD	<1 cup/day	≥1 cup/day	*p*-Value
Number (%)	*n* = 13,439	*n* = 4515	*n* = 5118		*n* = 4879	*n* = 1641	*n* = 1821		*n* = 8560	*n* = 2874	*n* = 3297	
Age (years), medium (min–max)	57 (32–75)	56 (32–76)	53 (32–76)	<0.001 *	57 (32–75)	57 (32–76)	54 (32–76)	<0.001	57 (32–75)	56 (32–76)	52 (32–75)	<0.001
Age ≥ 65 years	2952 (22.0%)	881 (19.5%)	709 (13.9%)	<0.001 *	1235 (25.3%)	377 (23.0%)	355 (19.5%)	<0.001 *	1717 (20.1%)	504 (17.5%)	354 (10.7%)	<0.001 *
Gender (male)	4879 (36.3%)	1641 (36.3%)	1821 (35.6%)	0.627								
Menopause									5751 (67.2%)	1869 (65.0%)	1777 (53.9%)	<0.001 *
BMI (kg/m^2^)				<0.001 *				0.001 *				0.004 *
BMI < 24 kg/m^2^	7034 (52.3%)	2254 (49.9%)	2466 (48.2%)		1912 (39.2%)	589 (35.9%)	615 (33.8%)		5122 (59.8%)	1665 (57.9%)	1851 (56.1%)	
BMI 24–27 kg/m^2^	3773 (28.1%)	1342 (29.7%)	1529 (29.9%)		1746 (35.8%)	632 (38.5%)	700 (38.4%)		2027 (23.7%)	710 (24.7%)	829 (25.1%)	
BMI > 27 kg/m^2^	2632 (19.6%)	919 (20.4%)	1123 (21.9%)		1221 (25.0%)	420 (25.6%)	506 (27.8%)		1411 (16.5%)	499 (17.4%)	617 (18.7%)	
Education level (collage or above)	6445 (48.0%)	2441 (54.1%)	2914 (56.9%)	<0.001 *	2869 (58.8%)	1119 (68.2%)	1231 (67.6%)	<0.001 *	3576 (41.8%)	1322 (46.0%)	1683 (51.0%)	<0.001 *
Marriage status (married)	10440 (777%)	3556 (78.8%)	3951 (77.2%)	0.167	4202 (86.1%)	1462 (89.1%)	1572 (86.3%)	0.008 *	6238 (72.9%)	2094 (72.9%)	2379 (72.2%)	0.720
Occupational status (employed)	7334 (54.6%)	2636 (58.4%)	3332 (65.1%)	<0.001 *	1824 (37.4%)	543 (33.1%)	536 (29.4%)	<0.001 *	4279 (50.0%)	1538 (53.5%)	2047 (62.1%)	<0.001 *
Place of residence				0.002 *				0.001 *				0.452
Urban area	7655 (57.0%)	2595 (57.5%)	3078 (60.1%)		2671 (54.7%)	900 (54.8%)	1097 (60.2%)		4984 (58.2%)	1695 (59.0%)	1981 (60.1%)	
Suburban area	4929 (36.7%)	1625 (36.0%)	1754 (34.3%)		1849 (37.9%)	612 (37.3%)	615 (33.8%)		3080 (36.0%)	1013 (35.2%)	1139 (34.5%)	
Rural area	855 (6.4%)	295 (6.5%)	286 (5.6%)		359 (7.4%)	129 (7.9%)	109 (6.0%)		496 (5.8%)	166 (5.8%)	177 (5.4%)	
Monthly personal Income (TWD)				<0.001 *				<0.001 *				<0.001 *
0–20,000	3966 (29.5%)	1091 (24.2%)	1019 (19.9%)		833 (17.1%)	202 (12.3%)	206 (11.3%)		3133 (36.6%)	889 (30.9%)	813 (24.7%)	
20,001–40,000	4309 (32.1%)	1406 (31.1%)	1733 (33.9%)		1284 (26.3%)	360 (21.9%)	426 (23.4%)		3025 (35.3%)	1046 (36.4%)	1307 (39.6%)	
40,001 or more	5164 (38.4%)	2018 (44.7%)	2366 (46.2%)		2762 (56.6%)	1079 (65.8%)	1189 (65.3%)		2402 (28.1%)	939 (32.7%)	1177 (35.7%)	
Cigarette smoking status				<0.001 *				<0.001 *				<0.001 *
Nonsmoker	11129 (82.8%)	3624 (80.3%)	3955 (77.3%)		2821 (57.8%)	865 (52.7%)	870 (47.8%)		8308 (97.1%)	2759 (96.0%)	3085 (93.6%)	
Former smokers	1347 (10.0%)	581 (12.9%)	633 (12.4%)		1213 (24.9%)	516 (31.4%)	529 (29.0%)		134 (1.6%)	65 (2.3%)	104 (3.2%)	
Current smokers	963 (7.2%)	310 (6.9%)	530 (10.4%)		845 (17.3%)	260 (15.8%)	422 (23.2%)		118 (1.4%)	50 (1.7%)	108 (3.3%)	
Secondhand smoke exposure	1217 (9.1%)	400 (8.9%)	498 (9.7%)	0.264	570 (11.7%)	180 (11.0%)	216 (11.9%)	0.674	647 (7.6%)	220 (7.7%)	282 (8.6%)	0.185
Alcohol drinking status												
Nondrinkers	12128 (90.2%)	3993 (88.4%)	4463 (87.2%)	<0.001 *	3773 (77.3%)	1225 (74.6%)	1335 (73.3%)	<0.001 *	8355 (97.6%)	2768 (96.3%)	3128 (94.9%)	<0.001 *
Former drinkers	503 (3.7%)	151 (3.3%)	213 (4.2%)		413 (8.5%)	115 (7.0%)	166 (9.1%)		90 (1.1%)	36 (1.3%)	47 (1.4%)	
Current drinkers	808 (6.0%)	371 (8.2%)	442 (8.6%)		693 (14.2%)	301 (18.3%)	320 (17.6%)		115 (1.3%)	70 (2.4%)	122 (3.7%)	
Habitual Tea Drinking	2879 (21.4%)	1250 (27.7%)	1379 (26.9%)	<0.001 *	1539 (31.5%)	584 (35.6%)	589 (32.3%)	0.010 *	1340 (15.7%)	666 (23.2%)	790 (24.0%)	<0.001 *
Coffee Types				<0.001 *				<0.001 *				<0.001 *
Black coffee	0	2603 (57.7%)	2515 (49.1%)		0	1064 (64.8%)	1055 (57.9%)		0	1539 (53.5%)	1460 (44.3%)	
Coffee with creamer	0	622 (13.8%)	706 (13.8%)		0	209 (12.7%)	256 (14.1%)		0	413 (14.4%)	450 (13.6%)	
Coffee with milk	0	965 (21.4%)	1466 (28.6%)		0	272 (16.6%)	268 (20.2%)		0	693 (24.1%)	1098 (33.3%)	
Others	0	325 (7.2%)	431 (8.4%)		0	96 (5.9%)	142 (7.8%)		0	229 (8.0%)	289 (8.8%)	
Vegetarian	723 (5.4%)	185 (4.1%)	199 (3.9%)	<0.001 *	184 (3.8%)	53 (3.2%)	59 (3.2%)	0.427	539 (6.3%)	132 (4.6%)	140 (4.2%)	<0.001 *
Regular exercise	6425 (47.8%)	2276 (50.4%)	2328 (45.5%)	<0.001 *	2341 (48.0%)	862 (52.5%)	920 (50.5%)	0.004 *	4084 (47.7%)	1414 (49.2%)	1408 (42.7%)	<0.001 *
Daily sleep hours less than 6 h	2094 (15.6%)	598 (13.2%)	641 (12.5%)	<0.001 *	682 (14.0%)	175 (10.7%)	217 (11.9%)	0.001 *	1412 (16.5%)	423 (14.7%)	424 (12.9%)	<0.001 *
Metabolic syndrome parameter abnormal number and ratio												
Waist Circumference > 90 cm in men and >80 cm in women	6623 (49.3%)	2275 (50.4%)	2548 (49.8%)	0.420	1918 (39.3%)	648 (39.5%)	751 (41.2%)	0.345	4705 (55.0%)	1627 (56.6%)	1797 (54.5%)	0.207
Triglycerides > 150 mg/dL	3115 (23.2%)	1052 (23.3%)	1026 (20.0%)	<0.001 *	1475 (30.2%)	510 (31.1%)	509 (28.0%)	0.098	1640 (19.2%)	542 (18.9%)	517 (15.7%)	<0.001 *
HDL cholesterol < 40 mg/dL in men and <50 mg/dL in women	3707 (27.6%)	1160 (25.7%)	1149 (22.5%)	<0.001 *	1162 (23.8%)	369 (22.5%)	392 (21.5%)	0.117	2545 (29.7%)	791 (27.5%)	757 (23.0%)	<0.001 *
Systolic blood pressure > 130 mmHg or diastolic blood pressure > 85 mmHg or hypertensive treatment	5718 (42.5%)	1890 (41.9%)	1916 (37.4%)	<0.001 *	2643 (54.2%)	892 (54.4%)	965 (53.0%)	0.645	3075 (35.9%)	998 (34.7%)	951 (28.8%)	<0.001 *
Fasting plasma glucose > 100 mg/dL or diabetes treatment	3423 (25.5%)	1162 (25.7%)	1206 (23.6%)	0.015 *	1589 (32.6%)	558 (34.0%)	583 (32.0%)	0.430	1834 (21.4%)	604 (21.0%)	623 (18.9%)	0.009 *

* *p* < 0.05. Min, minimum. Max, maximum. NCD, non-coffee drinker. BMI, body mass index. TWD, New Taiwan dollar.

**Table 2 nutrients-14-01867-t002:** The unadjusted odds ratios of daily coffee drinking volume and general characteristics in metabolic syndrome in Taiwanese participants aged 30–79 years.

Index	OR (95% CI)	Overall *p*-Value	OR (95% CI)	Male *p*-Value	OR (95% CI)	Female *p*-Value
Coffee drinking volume (1 cup = 8 oz)						
Non-coffee drinker	1.00		1.00		1.00	
<1 cup/day	0.94 (0.87–1.02)	0.131	1.01 (0.89–1.14)	0.924	0.90 (0.82–1.00)	0.044 *
≥1 cup/day	0.82 (0.76–0.89)	<0.001 *	0.93 (0.83–1.05)	0.257	0.76 (0.69–0.83)	<0.001 *
Age ≥ 65 years	1.83 (1.71–1.97)	<0.001 *	1.26 (1.13–1.40)	<0.001 *	2.32 (2.12–2.55)	<0.001 *
BMI (kg/m^2^)						
BMI < 24 kg/m^2^	1.00		1.00		1.00	
BMI 24–27 kg/m^2^	4.247 (3.92–4.60)	<0.001 *	4.52 (3.89–5.25)	<0.001 *	4.54 (4.12–5.00)	<0.001 *
BMI > 27 kg/m^2^	11.96 (11.00–13.00)	<0.001 *	17.51 (15.00–20.45)	<0.001 *	9.80 (8.84–10.88)	<0.001 *
Education level (collage or above)	0.61 (0.57–0.64)	0.000 *	0.67 (0.61–0.74)	0.000 *	0.51 (0.47–0.55)	0.000 *
Marriage status (married)	0.91 (0.85–0.98)	0.012 *	0.93 (0.81–1.07)	0.326	0.84 (0.78–0.92)	0.000 *
Occupational status (employed)	0.69 (0.65–0.73)	0.000 *	0.86 (0.78–0.94)	0.002 *	0.57 (0.53–0.62)	0.000 *
Place of residence						
Urban area	1.00		1.00		1.00	
Suburban area	1.03 (0.97–1.10)	0.336	1.04 (0.94–1.14)	0.50	1.02 (0.94–1.11)	0.638
Rural area	0.95 (0.83–1.07)	0.380	0.82 (0.67–0.99)	0.045 *	1.02 (0.86–1.21)	0.799
Monthly personal income (TWD)						
0–20,000	1.00		1.00		1.00	
20,001–40,000	0.73 (0.68–0.79)	<0.001 *	0.82 (0.70–0.95)	0.008 *	0.68 (0.62–0.74)	<0.001 *
40,001 or more	0.69 (0.65–075)	<0.001 *	0.71 (0.62–0.81)	<0.001 *	0.57 (0.52–0.63)	<0.001 *
Cigarette smoking status						
Non-smoker	1.00		1.00		1.00	
Former smokers	1.44 (1.32–1.58)	<0.001 *	1.46 (1.31–1.63)	<0.001 *	0.90 (0.69–1.19)	0.456
Current smokers	1.60 (1.44–1.77)	<0.001 *	1.62 (1.43–1.83)	<0.001 *	1.12 (0.85–1.47)	0.413
Secondhand smoke exposure	1.17 (1.06–1.29)	0.002 *	1.35 (1.17–1.56)	<0.001 *	0.97 (0.84–1.12)	0.648
Alcohol drinking status						
Nondrinkers	1.00		1.00		1.00	
Former drinkers	1.72 (1.49–1.98)	<0.001 *	1.63 (1.39–1.92)	<0.001 *	1.44 (1.04–1.99)	0.028 *
Current drinkers	1.41 (1.27–1.58)	<0.001 *	1.48 (1.30–1.68)	<0.001 *	0.69 (0.52–0.93)	0.014 *
Habitual tea drinking	1.20 (1.12–1.28)	0.000 *	1.15 (1.04–1.27)	0.008 *	1.15 (1.05–1.26)	0.004 *
Regular exercise	1.02 (0.96–1.08)	0.542	0.81 (0.74–0.89)	0.000 *	1.17 (1.08–1.26)	0.000 *
Vegetarian	1.12 (0.98–1.29)	0.088	1.05 (0.82–1.36)	0.692	1.20 (1.02–1.41)	0.025 *
Daily sleep hours less than 6 h	0.90 (0.83–0.98)	0.014 *	0.93 (0.81–1.07)	0.320	0.87 (0.78–0.96)	0.007 *
Menopause (female)					0.36 (0.33–0.40)	0.000 *

* *p* < 0.05. NCD, non-coffee drinker. BMI, body mass index. TWD, New Taiwan dollar. oz, ounce.

**Table 3 nutrients-14-01867-t003:** The adjusted odds ratio of different daily coffee drinking volume in metabolic parameters in Taiwanese participants aged 30–79 years.

	Overall			Male			Female		
Index	NCD	<1 cup/day	≥1 cup/day	NCD	<1 cup/day	≥1 cup/day	NCD	<1 cup/day	≥1 cup/day
No. of participants	13,439	4515	5118	4879	1641	1821	8560	2874	3297
MetS									
No. of cases	3607	1160	1184	1430	483	508	2177	677	676
Model 1: AOR (95% CI)	Ref.	0.90 (0.82–0.98)	0.80 (0.73–0.87)	Ref.	0.97 (0.84–1.11)	0.87 (0.76–1.00)	Ref.	0.86 (0.77–0.96)	0.77 (0.69–0.87)
*p*-value		0.15	<0.001 *		0.634	0.05		0.007 *	<0.001 *
Model 2: AOR (95% CI)	Ref.	0.92 (0.84–0.99)	0.80 (0.73–0.87)	Ref.	1.00 (0.87–1.16)	0.87 (0.76–0.99)	Ref.	0.86 (0.77–0.97)	0.78 (0.69–0.87)
*p*-value		0.048	<0.001 *		0.954	0.047 *		0.01 *	<0.001 *
Abnormal Waist									
No. of cases	6623	2275	2548	1918	648	751	4705	1627	1797
Model 1: AOR (95% CI)	Ref.	0.92 (0.84–1.01)	0.87 (0.79–0.95)	Ref.	0.91 (0.77–1.07)	0.92 (0.79–1.08)	Ref.	0.94 (0.84–1.05)	0.86 (0.77–0.96)
*p*-value		0.087	0.003 *		0.267	0.319		0.274	0.007 *
Model 2: AOR (95% CI)	Ref.	0.92 (0.84–1.02)	0.86 (0.79–0.95)	Ref.	0.91 (0.77–1.08)	0.86 (0.76–1.05)	Ref.	0.94 (0.84–1.06)	0.86 (0.77–0.96)
*p*-value		0.103	0.002 *		0.272	0.180		0.293	0.008 *
Abnormal TG									
No. of cases	3115	1052	1026	1475	510	509	1640	542	517
Model 1: AOR (95% CI)	Ref.	0.99 (0.91–1.07)	0.80 (0.74–0.87)	Ref.	1.01 (0.89–1.14)	0.82 (0.73–0.93)	Ref.	0.97 (0.87–1.09)	0.81 (0.73–0.91)
*p*-value		0.722	<0.001 *		0.915	0.002 *		0.594	<0.001 *
Model 2: AOR (95% CI)	Ref.	1.00 (0.92–1.09)	0.80 (0.73–0.87)	Ref.	1.02 (0.90–1.16)	0.81 (0.72–0.92)	Ref.	0.97 (0.87–1.09)	0.80 (0.72–0.90)
*p*-value		0.981	<0.001 *		0.731	0.001 *		0.647	<0.001 *
Abnormal HDL-c									
No. of cases									
Model 1: AOR (95% CI)	Ref.	0.88 (0.81–0.95)	0.71 (0.66–0.77)	Ref.	0.90 (0.78–1.03)	0.84 (0.73–0.96)	Ref.	0.87 (0.79–0.96)	0.66 (0.60–0.72)
*p*-value		0.001 *	<0.001 *		0.131	0.009 *		0.004 *	<0.001 *
Model 2: AOR (95% CI)	Ref.	0.90 (0.83–0.98)	0.72 (0.66–0.78)	Ref.	0.96 (0.83–1.10)	0.83 (0.72–0.95)	Ref.	0.88 (0.80–0.98)	0.66 (0.60–0.73)
*p*-value		0.014 *	<0.001 *		0.530	0.007 *		0.014 *	<0.001 *
Abnormal BP									
No. of cases	5718	1890	1916	2643	892	965	3075	998	951
Model 1: AOR (95% CI)	Ref.	0.96 (0.89–1.04)	0.91 (0.85–0.99)	Ref.	0.99 (0.88–1.12)	1.02 (0.91–1.11)	Ref.	0.94 (0.85–1.04)	0.86 (0.78–0.95)
*p*-value		0.282	0.018 *		0.903	0.747		0.218	0.004 *
Model 2: AOR (95% CI)	Ref.	0.96 (0.89–1.04)	0.92 (0.85–0.99)	Ref.	0.98 (0.87–1.11)	1.01 (0.90–1.14)	Ref.	0.95 (0.86–1.05)	0.87 (0.79–0.97)
*p*-value		0.316	0.025 *		0.794	0.822		0.278	0.008 *
Abnormal FPG									
No. of cases	3423	1162	1206	1589	558	583	1834	604	623
Model 1: AOR (95% CI)	Ref.	1.02 (0.94–1.10)	1.01 (0.93–1.09)	Ref.	1.07 (0.95–1.22)	1.04 (0.92–1.18)	Ref.	0.98 (0.88–1.09)	0.99 (0.89–1.10)
*p*-value		0.704	0.876		0.266	0.519		0.669	0.858
Model 2: AOR (95% CI)	Ref.	1.02 (0.94–1.10)	1.00 (0.92–1.09)	Ref.	1.09 (0.96–1.24)	1.05 (0.93–1.19)	Ref.	0.96 (0.86–1.07)	0.97 (0.87–1.09)
*p*-value		0.703	0.998		0.165	0.439		0.490	0.604

* *p* < 0.05. Ref., reference. MetS: metabolic syndrome; NCD: non-coffee drinker; abnormal waist: waist circumference > 90 cm in men and >80 cm in women; abnormal TG: triglycerides > 150 mg/dL; abnormal HDL-c: HDL cholesterol < 40 mg/dL in men and <50 mg/dL in women; abnormal BP: systolic blood pressure > 130 mmHg or diastolic blood pressure > 85 mmHg or hypertensive treatment; abnormal FPG: fasting plasma glucose > 100 mg/dL or diabetes treatment. Model 1: reference = non-coffee consumer, adjusted for gender (for all participants), age, and body mass index. Model 2: reference = non-coffee consumer, adjusted for gender (for all participants), age, BMI, education level, marriage status, occupational status, place of residence, cigarette smoking status, secondhand smoke exposure, alcohol drinking, habitual tea drinking, regular exercise, vegetarian, daily sleep hours (less than 6 h), monthly income, and menopause (for female).

**Table 4 nutrients-14-01867-t004:** The adjusted odds ratio of different daily coffee drinking volume in metabolic syndrome in three coffee types in Taiwanese participants aged 30–79 years.

	Overall			Male			Female		
Index	NCD	<1 cup/day	≥1 cup/day	NCD	<1 cup/day	≥1 cup/day	NCD	<1 cup/day	≥1 cup/day
Group 1:	NCD	BC < 1 cup/day	BC ≥ 1 cup/day	NCD	BC < 1 cup/day	BC ≥ 1 cup/day	NCD	BC < 1 cup/day	BC ≥ 1 cup/day
No. of participants	13,439	2603	2515	4879	1064	1055	8560	1539	1460
No. of cases	3607	692	617	1430	324	289	2177	368	328
Model 1: AOR (95% CI)	Ref.	0.90 (0.81–1.00)	0.80 (0.72–0.90)	Ref.	1.01 (0.86–1.19)	0.82 (0.69–0.97)	Ref.	0.83 (0.72–0.96)	0.80 (0.69–0.93)
*p*-value		0.051	<0.001 *		0.937	0.023 *		0.009 *	0.004 *
Model 2: AOR (95% CI)	Ref.	0.93 (0.83–1.03)	0.80 (0.72–0.90)	Ref.	1.07 (0.90–1.26)	0.82 (0.69–0.98)	Ref.	0.84 (0.73–0.96)	0.81 (0.69–0.94)
*p*-value		0.156	<0.001 *		0.460	0.026 *		0.014 *	0.006 *
Group 2:	NCD	CC < 1 cup/day	CC ≥ 1 cup/day	NCD	CC < 1 cup/day	CC ≥ 1 cup/day	NCD	CC < 1 cup/day	CC ≥ 1 cup/day
No. of participants	13,439	622	706	4879	209	256	8560	413	450
No. of cases	3607	188	185	1430	65	72	2177	123	113
Model 1: AOR (95% CI)	Ref.	0.95 (0.78–1.16)	0.78 (0.64–0.95)	Ref.	0.86 (0.61–1.22)	0.88 (0.64–1.21)	Ref.	0.98 (0.76–1.26)	0.76 (0.59–0.98)
*p*-value		0.621	0.015 *		0.406	0.431		0.855	0.034 *
Model 2: AOR (95% CI)	Ref.	0.93 (0.76–1.13)	0.75 (0.62–0.92)	Ref.	0.85 (0.60–1.21)	0.82 (0.60–1.14)	Ref.	0.95 (0.74–1.23)	0.74 (0.57–0.95)
*p*-value		0.457	0.005 *		0.371	0.238		0.716	0.019 *
Group 3:	NCD	CM < 1 cup/day	CM ≥ 1 cup/day	NCD	CM < 1 cup/day	CM ≥ 1 cup/day	NCD	CM < 1 cup/day	CM ≥ 1 cup/day
No. of participants	13,439	965	1466	4879	272	368	8560	693	1098
No. of cases	3607 (26.8%)	220	285	1430	73	105	2177	147 (21.2%)	180
Model 1: AOR (95% CI)	Ref.	0.98 (0.82–1.17)	0.76 (0.66–0.89)	Ref.	1.02 (0.75–1.41)	1.00 (0.76–1.31)	Ref.	0.96 (0.78–1.19)	0.70 (0.58–0.84)
*p*-value		0.809	0.001 *		0.884	0.978		0.716	<0.001 *
Model 2: AOR (95% CI)	Ref.	1.01 (0.85–1.20)	0.78 (0.67–0.91)	Ref.	1.08 (0.78–1.49)	1.02 (0.77–1.34)	Ref.	0.98 (0.79–1.21)	0.71 (0.59–0.86)
*p*-value		0.924	0.002 *		0.644	0.903		0.836	<0.001 *

* *p* < 0.05. Ref., reference. MetS, Metabolic syndrome; NCD, non-coffee drinker; BC, black coffee; CC, coffee with creamer; CM, coffee with milk. Group 1: the population of non-coffee drinkers and black coffee drinkers; Group 2: the population of non-coffee drinkers and black coffee with cream drinkers; Group 3: the population of non-coffee drinkers and black coffee with milk drinkers; Group4: the population of non-coffee drinkers and other drinkers. Model 1: reference = non-coffee consumer, adjusted for gender (for all participants), age, and body mass index. Model 2: reference = non-coffee consumer, adjusted for gender (for all participants), age, BMI, education level, marriage status, occupational status, place of residence, cigarette smoking status, secondhand smoke exposure, alcohol drinking, habitual tea drinking, regular exercise, vegetarian, daily sleep hours (less than 6 h), monthly income, and menopause (for female).

## Data Availability

Not applicable.

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
