# Peer review of "The Relationship between Habitual Coffee Drinking and the Prevalence of Metabolic Syndrome in Taiwanese Adults: Evidence from the Taiwan Biobank Database"

_nutrients, 2022, doi:10.3390/nu14091867_

Round 1

Reviewer 1 Report

Dear Authors,

Thank you for your manuscript. Please consider my comments below.

Please list coffee types in the abstract and provide criteria for defining metabolic syndrome (MS). Please indicate the age of the study participants in the abstract (mean and range age for men and women).

In the Introduction, lines 40-43, the elevated level of triglycerides is missing when defining criteria for the MS.

I recommend bettering the structure of the Methods section as it is messy and confusing. Coffee consumption is described together with the covariates but is a major study variable. Please move it into a separate subsection. Also, in tables, I can see a coffee type as "other", but from the Methods section, it is not clear what was classified as other.

Please move the description of the components of the MS to subsection 2.2. and provide references for the biochemical analyses.

Please separate the description of other covariates in a new paragraph and provide references. Make sure that the description of the covariates is complete, e.g., vegetarian diet there was only mentioned, but it was not explained what were the criteria for classifying subjects as vegetarians (there are different types of vegetarian diets). Also, a food frequency questionnaire is mentioned in line 93, but it is not clear if the adjustment was made for dietary habits or if only a coffee consumption was used for this study?

Please move the categorisation criteria into BMI groups to the description of anthropometric measurements and provide a reference.

Author Response

Response to Reviewer 1 Comments

Point 1: Please list coffee types in the abstract and provide criteria for defining metabolic syndrome (MS).

Response 1: Thanks a lot for your advice! We have listed the coffee type and criteria of metabolic synrome in line 22-23.

Abstract:

Previous studies revealed inconsistent results between coffee drinking and metabolic syndrome (MetS). We evaluated the effects of habitual coffee drinking on the prevalence of MetS among men and women. We conducted a nationwide, cross-sectional study using 23,073 adults obtained from the Taiwan Biobank database (mean±SD (range) age, 54.57±0.07 (30–79) years; 8,341 men and 14,731 [63.8%] women). Adults who drank more than one cup of coffee per day (n=5,118) and those who drank less than a cup per day (n=4,515) were compared with none-coffee drinkers (n=13,439). Multivariate logistic regression models were used to evaluate the risk of MetS between the two groups. Separate models were also estimated for sex-stratified and habitual coffee-type-stratified (black coffee [BC], coffee with creamer [CC], and coffee with milk [CM]) subgroup analyses. The MetS diagnosis was based on at least three of the five metabolic abnormalities.

 Point 2: Please indicate the age of the study participants in the abstract (mean and range age for men and women).

Response 2: Thanks a lot for your advice! We have indicated the age of the study participants in line 17-18.

Abstract:

Previous studies revealed inconsistent results between coffee drinking and metabolic syndrome (MetS). We evaluated the effects of habitual coffee drinking on the prevalence of MetS among men and women. We conducted a nationwide, cross-sectional study using 23,073 adults obtained from the Taiwan Biobank database (mean±SD (range) age, 54.57±0.07 (30–79) years; 8,341 men and 14,731 [63.8%] women).

Point 3: In the Introduction, lines 40-43, the elevated level of triglycerides is missing when defining criteria for the MS.

Response 3: Thanks a lot for your advice! We have corrected the criteria in line 42-45.

 The Metabolic syndrome (MetS) is defined as a metabolic disorder, which is character-ized by high fasting blood glucose, abdominal obesity, abnormal high-density lipopro-tein cholesterol (HDL-C), high triglycerides, and high blood pressure, that is the hall-mark of insulin resistance.

 Point 4: I recommend bettering the structure of the Methods section as it is messy and confusing. Coffee consumption is described together with the covariates but is a major study variable. Please move it into a separate subsection.

Response 4: Thanks a lot for your advice! We have moved the coffee consumption into “2.3. Habitual Coffee drinking”, which was in line 88-100.

2.3. Habitual Coffee drinking

Habitual coffee drinking questionnaire including daily coffee intake volume and habitual coffee type. Participants were defined as coffee drinkers if they drank coffee three or more times per week [40]. Based on the mean daily intake volume (where 1 cup of coffee = 236.59 mL [8 fluid ounces]) [41], the coffee-drinking behavior of the partici-pants was classified under three categories: no habitual coffee drinking, less than one cup of coffee per day, and at least one cup of coffee per day. Habitual coffee type in-cluded three types of coffee (black coffee, coffee with creamer, and coffee with milk), which are common coffee types that are consumed by the Taiwanese people. Black coffee was defined as coffee powder or extracts, potentially with added water but no creamer or milk. Coffee with creamer and coffee with milk were defined as black coffee added creamer and milk, respectively. Each participant who was defined as coffee drinker must record the coffee type. The subject who has more than one habitual coffee type was categories as others.

 Point 5: Also, in tables, I can see a coffee type as "other", but from the Methods section, it is not clear what was classified as other.

Response 5: Thanks a lot for your advice! We have explained the coffee type “others” in line 100.

 2.3. Habitual Coffee drinking

Habitual coffee drinking questionnaire including daily coffee intake volume and habitual coffee type. Participants were defined as coffee drinkers if they drank coffee three or more times per week [40]. Based on the mean daily intake volume (where 1 cup of coffee = 236.59 mL [8 fluid ounces]) [41], the coffee-drinking behavior of the partici-pants was classified under three categories: no habitual coffee drinking, less than one cup of coffee per day, and at least one cup of coffee per day. Habitual coffee type in-cluded three types of coffee (black coffee, coffee with creamer, and coffee with milk), which are common coffee types that are consumed by the Taiwanese people. Black coffee was defined as coffee powder or extracts, potentially with added water but no creamer or milk. Coffee with creamer and coffee with milk were defined as black coffee added creamer and milk, respectively. Each participant who was defined as coffee drinker must record the coffee type. The subject who has more than one habitual coffee type was categories as others.

 Point 6: Please move the description of the components of the MS to subsection 2.2. and provide references for the biochemical analyses.

Response 6: Thanks a lot for your advice! We have moved the description of the components of the MS to subsection 2.2 in line 78-80.

2.2. Definition of Metabolic Syndrome

The waist circumference was determined through physical examination. Fasting blood sugar, serum creatinine, HDL-C, low-density lipoprotein cholesterol (LDL-C), and tri-glyceride levels were determined through biochemical examinations. Based on the modified National Cholesterol Education Program Adult Treatment Panel III (NCEP ATP III) definition, MetS is diagnosed if at least three of the following five metabolic abnormalities are present: (1) waist circumference ≥90 cm for men and ≥80 cm for women; (2) triglycerides ≥150 mg/dL or pharmacotherapy for hyperlipidemia; (3) HDL cholesterol <40 mg/dL for men and <50 mg/dL for women; (4) blood pressure ≥130/85 mmHg or pharmacotherapy for hypertension; and/or (5) fasting blood glucose ≥100 mg/dL or taking glucose-lowering medication [18].

 Point 7: Please separate the description of other covariates in a new paragraph and provide references.

Response 7: Thanks a lot for your advice! We have moved the “other covariates” into paragraphy 2.4, which was in line 101-128.

 Point 8: Make sure that the description of the covariates is complete, e.g., vegetarian diet there was only mentioned, but it was not explained what were the criteria for classifying subjects as vegetarians (there are different types of vegetarian diets).

Response 8: Thanks a lot for your advice! We have explained the criteria of vegetarian diet in line 116-118.

 2.4. Other Covariates

Vegetarians were those who maintain a daily vegetarian diet (abstaining from the consumption of red meat, poultry, seafood, insects, and the flesh of any other animal) for at least 6 months before the data collection date. Habitual tea drinkers were those who drank tea at least once a day [40].

 Point 9: Also, a food frequency questionnaire is mentioned in line 93, but it is not clear if the adjustment was made for dietary habits or if only a coffee consumption was used for this study?

Response 9: Thanks a lot for your advice! The questionnaire of coffe drinking type and inatke volume form Taiwan biobank was refer to “Nutrition and Health Survey in Taiwan (NAHSIT) 1993-1996: Dietary Nutrient Intakes Assessed by 24-Hour Recall”, however, in this study, the questionnaire was just for survey of coffee type and daily intake volume, not for diatery adjustement. Therefore, It was indeed not appropriate mentioned as “food frequency questionnaire”, but only a questionnaire. I has corrected it in line 89-90.

2.3. Habitual Coffee drinking

Habitual coffee drinking questionnaire including daily coffee intake volume and ha-bitual coffee type. Participants were defined as coffee drinkers if they drank coffee three or more times per week [40]. Based on the mean daily intake volume (where 1 cup of coffee = 236.59 mL [8 fluid ounces]) [41], the coffee-drinking behavior of the participants was clas-sified under three categories: no habitual coffee drinking, less than one cup of coffee per day, and at least one cup of coffee per day. Habitual coffee type included three types of coffee (black coffee, coffee with creamer, and coffee with milk), which are common coffee types that are consumed by the Taiwanese people. Black coffee was defined as coffee powder or extracts, potentially with added water but no creamer or milk. Coffee with creamer and coffee with milk were defined as black coffee added creamer and milk, re-spectively. Each participant who was defined as coffee drinker must record the coffee type. The subject who has more than one habitual coffee type was categories as others.

 Point 10: Please move the categorisation criteria into BMI groups to the description of anthropometric measurements and provide a reference.

Response 10: Thanks a lot for your advice! We have corrected it in line 106-109.

 2.4. Covariates

Data on age, sex, education level, marital status, occupational status, residential area, monthly income, habitual tea drinking, alcohol consumption, regular exercise, smoking status, second-hand smoke exposure, daily sleep duration (hours), and menstrual status, were collected through questionnaires.

The body weight, and body height were determined through physical examination, and the body mass index (BMI) was calculated as the weight (in kg) divided by the height (in m2). BMI categories include underweight–normal, overweight, and obese (BMI <24.0, 24.0–26.9, and 27.0 kg/m2, respectively) [39].

Reviewer 2 Report

The manuscript entitled „The Relationship Between Habitual Coffee Drinking and the Prevalence of Metabolic Syndrome in Taiwanese Adults: Evidence from the Taiwan Biobank Database” presents interesting issue, but some issues should be corrected.

Abstract:

Brief justification of the study should be presented.

Aim of the study should be presented

Authors should not present too general conclusions – the conclusions should be directly associated with the conducted study and obtained results.

Introduction:

Authors present a number of basic and even very trivial information that should not be presented in a scientific manuscript (e.g. “Coffee is a brewed beverage that is prepared from roasted coffee beans”) – Authors should be aware that they do not prepare the basic manual for students, or column of the newspaper, but a scientific paper that should be interesting for researchers from the area of food and nutritional sciences, so they should understand that their readers will have the nutritional knowledge.

Authors should present what is already known and what are the “gaps” in the scientific knowledge to formulate the aim of their study.

Materials and Methods:

There is a really major problem associated with the fact that to assess intake, Authors applied a questionnaire that was not validated. It is very serious problem, as without properly conducted validation, we cannot state that this questionnaire is able to measure any variable. In general, nutritional studies can not be considered as valid and reliable if they use a questionnaire that previously was not validated.

At the same time, there are specific criteria of validation of FFQs, as described by Cade, so without such validation, intake can not be interpreted. Authors presented some results of intake, but they did not conduct a proper procedure of validation (including assessment of reproducibility and validity, as recommended by Cade).

It seems that Authors did not verify the normality of distribution of their data – they should do it and present the related methodology.

After verifying the normality of distribution, in case of parametric distribution mean ± SD should be presented, while for nonparametric distribution – median accompanied by minimum and maximum value.

The applied statistical test should be based on distribution

Results:

Authors should clearly present their results, as we even do not know if distribution is parametric or not

After verifying the normality of distribution, in case of parametric distribution mean ± SD should be presented, while for nonparametric distribution – median accompanied by minimum and maximum value.

The applied statistical test should be based on distribution

Discussion:

Authors should not reproduce the results within this section, but they should focus on the discussion.

Conclusions:

Authors should not present too general conclusions – the conclusions should be directly associated with the conducted study and obtained results.

Author Contributions:

This section should be properly presented.

Author Response

Response to Reviewer 2 Comments

Abstract:

Point 1: Brief justification of the study should be presented.

Response 1: Thanks a lot for your advice! We have add brief justification of the study in line 14-15.

Abstract:

Previous studies revealed inconsistent results between coffee drinking and metabolic syndrome (MetS). We evaluated the effects of habitual coffee drinking on the prevalence of MetS among men and women.

 Point 2: Aim of the study should be presented.

Response 2: Thanks a lot for your advice! We have add brief justification of the study in line 15-16.

Abstract:

Previous studies revealed inconsistent results between coffee drinking and metabolic syndrome (MetS). We evaluated the effects of habitual coffee drinking on the prevalence of MetS among men and women. We conducted a nationwide, cross-sectional study using 23,073 adults obtained from the Taiwan Biobank database (mean±SD (range) age, 54.57±0.07 (30–79) years; 8,341 men and 14,731 [63.8%] women).

Point 3: Authors should not present too general conclusions – the conclusions should be directly associated with the conducted study and obtained results

Response 3: Thanks a lot for your advice! We have correcred the conclusions in line 30-34.

Abstract: Previous studies revealed inconsistent results between coffee drinking and metabolic syndrome (MetS). We evaluated the effects of habitual coffee drinking on the prevalence of MetS among men and women. We conducted a nationwide, cross-sectional study using 23,073 adults obtained from the Taiwan Biobank database (mean±SD (range) age, 54.57±0.07 (30–79) years; 8,341 men and 14,731 [63.8%] women). Adults who drank more than one cup of coffee per day (n=5,118) and those who drank less than a cup per day (n=4,515) were compared with none-coffee drinkers (n=13,439). Multivariate logistic regression models were used to evaluate the risk of MetS between the two groups. Separate models were also estimated for sex-stratified and habitual coffee-type-stratified (black coffee [BC], coffee with creamer [CC], and coffee with milk [CM]) subgroup analyses. The MetS diagnosis was based on at least three of the five metabolic abnormalities. Coffee drinkers (≥1 cup/day) had a significantly lower prevalence of MetS than non-drinkers (AOR [95%CI]: 0.80 [0.73–0.87]). Women who drank any amount of coffee and types of coffee were more likely to have a significantly lower prevalence of MetS than non-drinkers. Only men who drank more than a cup of coffee per day or black coffee drinkers were more likely to have a lower prevalence of MetS. Our study results indicate that adults with habitual coffee drinking behaviors of more than a cup per day were associated with a lower prevalence of MetS. Even more, women could benefit from habitual coffee drinking of all three coffee types, whereas men could only benefit from drinking BC.

 Introduction:

Point 4: Authors present a number of basic and even very trivial information that should not be presented in a scientific manuscript (e.g. “Coffee is a brewed beverage that is prepared from roasted coffee beans”) – Authors should be aware that they do not prepare the basic manual for students, or column of the newspaper, but a scientific paper that should be interesting for researchers from the area of food and nutritional sciences, so they should understand that their readers will have the nutritional knowledge.

Response 4: Thanks a lot for your advice! We have deleted the sentence “Coffee is a brewed beverage that is prepared from roasted coffee beans”.

 Point 5: Authors should present what is already known and what are the “gaps” in the scientific knowledge to formulate the aim of their study.

Response 5: Thanks a lot for your advice! We have revised the whole content of introduction.

Introduction

Coffee has become increasingly popular in Taiwan in the past few decades. Coffee contains nearly 1000 components, such as caffeine, phenols (which contain chlorogenic acid and caffeic acid), and diterpenes (comprising cafestol and kahweol, lactones, niacin, and trigonelline) [1-3]. Many studies have proved that coffee consumption has a positive effect on chronic diseases. Habitual coffee drinking reduced arterial stiffness, blood pres-sure, and the incidence of new-onset hypertension [4-6], had a beneficial effect on weight loss [7-9], reduced intestinal glucose absorption [10], and decreased the incidence of type 2 diabetes mellitus [11, 12]. The Metabolic syndrome (MetS) is defined as a metabolic disor-der, which is characterized by high fasting blood glucose, abdominal obesity, abnormal high-density lipoprotein cholesterol (HDL-C), high triglycerides, and high blood pressure, that is the hallmark of insulin resistance. Individuals with MetS have a high risk of stroke and coronary heart disease [13, 14]. According to the survey of Nutrition and Health Sur-veys in Taiwan (NAHSIT), conducted during 1993-1996 and 2005-2008, a large growth on MetS prevalence was observed, from 13.6% to 25.5% [38]. Although coffee contains many components that are beneficial for individuals with chronic disease, previous studies of the relationship between habitual coffee drinking and MetS have generated inconsistent results, and even more among men and women. The inconsistent results might be generated from different study designs. Some studies focused on the influence of daily coffee intake volume, and some studies focused on habitual coffee type on the prevalence of MetS. Besides, same study results even generated from small sample size or a regional area, which was not adequate for presentation of general papulation [19-29, 32-36].

This study was conducted with an aim to evaluate the effect of habitual coffee drinking, particularly with regard to the daily intake volume and habitual coffee type, on the prevalence of MetS in the Taiwanese men and women through the Taiwan Biobank database.

 Materials and Methods:

Point 6: There is a really major problem associated with the fact that to assess intake, Authors applied a questionnaire that was not validated. It is very serious problem, as without properly conducted validation, we cannot state that this questionnaire is able to measure any variable. In general, nutritional studies can not be considered as valid and reliable if they use a questionnaire that previously was not validated.

Response 6: Thanks a lot for your advice! The questionnaire of coffe drinking type and inatke volume form Taiwan biobank was refer to “Nutrition and Health Survey in Taiwan (NAHSIT) 1993-1996: Dietary Nutrient Intakes Assessed by 24-Hour Recall”, however, in this study, the questionnaire was just for survey of coffee type and daily intake volume, not for diatery adjustement. Therefore, It was indeed not appropriate mentioned as “food frequency questionnaire”, but only a questionnaire. I has corrected it in line 89-90.

 Point 7: At the same time, there are specific criteria of validation of FFQs, as described by Cade, so without such validation, intake can not be interpreted. Authors presented some results of intake, but they did not conduct a proper procedure of validation (including assessment of reproducibility and validity, as recommended by Cade).

Response 7: Thanks a lot for your advice! The questionnaire of coffe drinking type and inatke volume form Taiwan biobank was refer to “Nutrition and Health Survey in Taiwan (NAHSIT) 1993-1996: Dietary Nutrient Intakes Assessed by 24-Hour Recall”, however, in this study, the questionnaire was just for survey of coffee type and daily intake volume, not for diatery adjustement. Therefore, It was indeed not appropriate mentioned as “food frequency questionnaire”, but only a questionnaire. I has corrected it in line 89-90.

 Point 8: It seems that Authors did not verify the normality of distribution of their data – they should do it and present the related methodology. After verifying the normality of distribution, in case of parametric distribution mean ± SD should be presented, while for nonparametric distribution – median accompanied by minimum and maximum value. The applied statistical test should be based on distribution.

Response 8: Thanks a lot for your advice! After verifying the normality of distribution, the continuous data (Age) was non-normal distribution (Kolmogorov-Smirnov test < 0.05), so it was presented as median accompanied by minimum and maximum value, and the variables in the different groups are compared using Kruskal-Wallis test.

 Results:

Point 9: Authors should clearly present their results, as we even do not know if distribution is parametric or not . After verifying the normality of distribution, in case of parametric distribution mean ± SD should be presented, while for nonparametric distribution – median accompanied by minimum and maximum value. The applied statistical test should be based on distribution.

Response 9: Thanks a lot for your advice! After verifying the normality of distribution, the continuous data (Age) was non-normal distribution (Kolmogorov-Smirnov test < 0.05), so it was presented as median accompanied by minimum and maximum value, and the variables in the different groups are compared using Kruskal-Wallis test. (p < 0.001 in Table1).

 Discussion:

Point 10: Authors should not reproduce the results within this section, but they should focus on the discussion.

Response 10: Thanks a lot for your advice! We have revised the whole content of discussion.

Discussion

According to our sex-stratified analysis, among the age group of 30-79 years, women who drank any amount of coffee and types of coffee were more likely to have a sig-nifi-cantly lower prevalence of MetS than non-drinkers. Besides, more daily coffee intake, women could get more benefit. Only men who drank more than a cup of coffee per day or black coffee drinkers were more likely to have a lower prevalence of MetS. Therefore, our study results indicated the habitual coffee drinking could protect from MetS in both Tai-wanese men and women, and women could benefit more than men. Among the compo-nents of MetS, both men and women (≥1 cup/day) had significantly lower prevalence of abnormal HDL-C and hypertriglyceridemia than non-drinker, which implied the poten-tial effect of habitual coffee drinking on lipid metabolism.

Worldwide, similar cross-sectional studies from Korea, Japan, Poland, Italy, and Denmark [16-31] have been conducted, with sample sizes ranging from hundreds to tens of thousands and with participants recruited from the regional to national levels. Our study data are from a large-scale, community-based cohort derived from the TWB data-base that comprises participants enrolled since 2008 from every administrative division of Taiwan. Thus, big data from our cross-sectional study among 23,072 adult Taiwanese participants that included 8,341 men and 14,731 women could provide solid evidence of a factual relationship. All the information from the questionnaire was collected from the in-terviews at the second visit, and is more reliable than information collected only from one visit.

Our study results showed the habitual coffee drinking could protect from MetS on both men and women. There are four cross-sectional studies that show different results from that of our study. Two cross-sectional studies with approximately 500 subjects re-cruited from a small community showed an insignificant association between coffee in-take and MetS [30-31]. Kim et al. showed a positive correlation between coffee drinking and MetS, but their participants were mainly instant coffee drinkers, and this differs from our study population [17]. Shin et al. enrolled only younger participants aged 30–40 years, which is a markedly different-aged population from that in our study, and showed insig-nificant relationship between coffee drinking and metabolic syndrome [21]. Many cross-sectional studies have shown similar results as in our study. Grosso et al. published a Polish arm of the Health, Alcohol and Psychosocial factors In Eastern Europe (HAPIEE) Study, which enrolled 8822 subjects from Poland, and showed a negative association be-tween high coffee intake and MetS among men and women [18].

Most of the cross-sectional studies focused on the association between the coffee in-take volume and MetS, rather than on the association between coffee types and MetS. Our study showed that women derive benefits from daily coffee intake volumes ≥1 cup/day for all three coffee types, but men can only derive this benefit with black coffee. A large, cross-sectional study in Korea. Kim et al. reported results from a 14,132-participant based Health Examinees (HEXA) study discussed the effect of both the daily coffee intake vol-ume and coffee types on MetS [26] and obtained similar results as our study, wherein an inverse correlation between daily coffee intake, of both coffee types, and MetS was ob-served among women.

The effect of coffee on the parameters of MetS from some cross-sectional studies were inconsistent [16, 18, 19, 29]. In our study, only the prevalence of FPG showed an insignifi-cant difference between the coffee drinkers and coffee non-drinkers. Because our study design only focused on the coffee intake volume and type, no further quantitative infor-mation about the addition of sugar, cream, and milk was obtained. Therefore, we could not determine why the results showed an insignificant correlation between coffee intake and the prevalence of abnormal fasting blood glucose. Meanwhile, the prevalence of ab-normal HDL-C was significantly lower than that of coffee non-drinkers in female partici-pants, regardless of the daily coffee intake volume, which implies that coffee strongly in-fluences the elevation of the serum HDL-C level in women. Another study that used data from the TWB revealed that coffee drinking was significantly associated with higher HDL-C levels in women, but not in men, even after adjusting for confounders including the hepatic lipase (LIPC) and cholesterol ester transfer protein (CETP) genes. Therefore, coffee drinking might be cardioprotective especially in women. [41].

There were at least three limitations in our study. First, despite a large database, our cross-sectional study design precluded the confirmation of causal associations. Second, because of the design of the questionnaire, we could not obtain quantitative information about added sugar, creamer, and milk. Third, our study enrolled only participants be-longing to the middle-aged to older adult population and, therefore, findings from the younger population could not be included in our results.

 Conclusions:

Point 11: Authors should not present too general conclusions – the conclusions should be directly associated with the conducted study and obtained results.

Response 11: Thanks a lot for your advice! We have revised the whole content of conclusion.

 Conclusions

Among the age group of 30-79 years, coffee drinking in excess of 1 cup per day could reduce the prevalence of MetS among Taiwanese men and women. Women could benefit from daily coffee intake volumes of more than 1 cup for all three coffee types, whereas men could only obtain a benefit with black coffee.

Author Contributions:

Point 12: This section should be properly presented.

Response 12: Thanks a lot for your advice! We have revised the content of auther contributions.

Conceptualization, M.Y.L., S.J.C.; formal analysis, M.Y.L.; writing—original draft preparation, M.Y.L.,H.Y.C.; writing—review and editing, M.Y.L., H.Y.C.; supervision, S.J.C., J.C.Y.L. All au-thors have read and agreed to the published version of the manuscript.

Round 2

Reviewer 1 Report

Dear Authors,

Congratulations and good job!

Author Response

Thanks a lot for your advice!

Reviewer 2 Report

The manuscript entitled „The Gender Effect on Relationship Between Habitual Coffee Drinking and the Prevalence of Metabolic Syndrome in Taiwanese Adults: Evidence from the Taiwan Biobank Database” presents interesting issue, but some issues should be corrected.

Abstract:

Aim of the study should be presented (e.g. “The aim of the study was…”).

Materials and Methods:

It seems that Authors did not verify the normality of distribution of their data – they should do it and present the related methodology (e.g. Shapiro-Wilk test).

Results:

Authors should clearly present their results, as we even do not know if distribution is parametric or not

Authors Contribution:

It seems that contribution of some Authors was only minor (SJC, JCYL) and they did not participate in preparing manuscript. There is a serious risk of a guest authorship procedure which is forbidden. In such case they should be rather presented in Acknowledgements Section and not be indicated as authors of the study.

Author Response

Response to Reviewer 2 Comments

Abstract

Point 1: Aim of the study should be presented (e.g. “The aim of the study was…”).

Response 1: Thanks a lot for your advice! We had revised the content of abstract.

Abstract:

Previous studies revealed inconsistent results between coffee drinking and metabolic syndrome (MetS). The aim of the study was to evaluate the effects of habitual coffee drinking on the prevalence of MetS among men and women.

Materials and Methods:

Point 2: It seems that Authors did not verify the normality of distribution of their data – they should do it and present the related methodology (e.g. Shapiro-Wilk test).

Response 2: Thanks a lot for your advice! We had revised the content of 2.5. Statistical Analysis.

2.5. Statistical Analysis

The characteristics of participants with different coffee-drinking behaviors were described as means and standard deviations or median accompanied by minimum and maximum value for continuous data and as the frequencies with percentages for categorical data. Each continuous variable was verified the normality of distribution by Shapiro-Wilk test. All variables in the different groups were compared using one-way analysis of variance (ANOVA) or Kruskal-Wallis test for continuous data, and the chi-square tests for categorical data.

Results:

Point 3: Authors should clearly present their results, as we even do not know if distribution is parametric or not

Response 3: Thanks a lot for your advice! We had revised the content of 3. Results.

 Results

Data were obtained from the TWB database for 23,072 adult Taiwanese (mean±SD (range) age, 54.57±0.07 (30–79) years; 8,341 male and 14,731 [63.8%] female [of whom 63.85% were menopausal]) participants. After verifying the normality of distribution, the continuous variable (Age) was non-parametric distribution. The comparison of the base-line characteristics of the non-coffee drinker, the coffee drinker consuming less than 1 cup/day, and the coffee drinker consuming ≥1 cup/day was shown in Table 1.

 Authors Contribution:

Point 4: It seems that contribution of some Authors was only minor (SJC, JCYL) and they did not participate in preparing manuscript. There is a serious risk of a guest authorship procedure which is forbidden. In such case they should be rather presented in Acknowledgements Section and not be indicated as authors of the study.

Response 4: Thanks a lot for your advice! We had revised the content of Authors Contribution.

Author Contributions: Conceptualization, M.Y.L., S.J.C.; formal analysis, M.Y.L., S.J.C.; writ-ing—original draft preparation, M.Y.L., H.Y.C.; writing—review and editing, M.Y.L., H.Y.C., S.J.C., J.C.Y.L.; supervision, S.J.C., J.C.Y.L. All authors have read and agreed to the published version of the manuscript.